# In Silico Analysis of Shiga Toxin-Producing *Escherichia coli* O157:H7 Strains from Presumptive Super- and Low-Shedder Cattle

**DOI:** 10.3390/toxins16020086

**Published:** 2024-02-05

**Authors:** Emmanuel W. Bumunang, Vinicius S. Castro, Trevor Alexander, Rahat Zaheer, Tim A. McAllister, Le Luo Guan, Kim Stanford

**Affiliations:** 1Agriculture and Agri-Food Canada, Lethbridge Research and Development Centre, Lethbridge, AB T1J 4B1, Canada; emmanuel.bumunang@agr.gc.ca (E.W.B.); trevor.alexander@agr.gc.ca (T.A.); rahat.zaheer@agr.gc.ca (R.Z.); tim.mcallister@agr.gc.ca (T.A.M.); 2Faculty of Agronomy and Zootechnics, Federal University of Mato Grosso (UFMT), Cuiabá 78010-715, Brazil; vinicius.castro1@ufmt.br; 3Department of Biological Sciences, University of Lethbridge, Lethbridge, AB T1K 1M4, Canada; 4Department of Agricultural, Food and Nutritional Science, University of Alberta, Edmonton, AB T6G 2P9, Canada; leluo.guan@ualberta.ca

**Keywords:** *Escherichia coli* O157:H7, super shedding, hybrid sequence analysis, Shiga toxin producing *E. coli*, comparative genomics

## Abstract

Cattle are the primary reservoir for STEC O157, with some shedding >10^4^ CFU/g in feces, a phenomenon known as super-shedding (SS). The mechanism(s) responsible for SS are not understood but have been attributed to the environment, host, and pathogen. This study aimed to compare genetic characteristics of STEC O157 strains from cattle in the same commercial feedlot pens with SS or low-shedding (LS) status. Strains from SS (n = 35) and LS (n = 28) collected from 11 pens in three feedlots were analyzed for virulence genes, Shiga toxin-carrying bacteriophage insertion sites, and phylogenetic relationships. In silico analysis showed limited variation regarding virulence gene profiles. *Stx*-encoding prophage insertion sites *mrlA* and *wrbA* for *stx1a* and *stx2a*, respectively, were all occupied, but two isolates had fragments of the *stx*-carrying phage in *mrlA* and *wrbA* loci without *stx1a* and *stx2a*. All strains screened for lineage-specific polymorphism assay (LSPA-6) were 111111, lineage I. Of the isolates, 61 and 2 were clades 1 and 8, respectively. Phylogenetic analysis revealed that pens with more than one SS had multiple distantly related clusters of SS and LS isolates. Although virulence genes and lineage were largely similar within and across feedlots, multiple genetic origins of strains within a single feedlot pen illustrate challenges for on-farm control of STEC.

## 1. Introduction

Shiga toxin-producing *Escherichia coli* O157:H7 (STEC O157) are important foodborne pathogens of concern to public health. A recent example is the STEC O157 outbreak in six locations of a Canadian daycare which recorded 349 lab-confirmed and 37 hospitalized cases [1]. Disease varied from a self-limiting intestinal pain and bloody diarrhea to severe hemolytic uremic syndrome [1]. The main virulence factors of STEC O157 include Shiga toxins 1 and 2 (*stx1*, *stx2*), intimin (*eae*), and the translocated intimin receptor (*tir*) genes, which are necessary for STEC O157 colonization in the host [2]. *Stx1* and *stx2* possess variant subtypes, *stx1a*, *stx1c*, *stx1d*, and *stx2a-stx2f* [3,4], with additional subtypes being discovered [5]. Shiga toxin genes are associated with a specific lamboid bacteriophage (phage) in STEC O157 [6] and *stx* subtypes may differ in clinical outcomes, alone or combined [7,8]. *Stx1a/2a* subtypes are commonly identified in STEC O157 isolated from cattle [9,10], the principal reservoir for STEC O157. Fecal contamination of food and water is the most common source of STEC O157 transmission [11] and it is estimated that approximately 20% of cattle shed > 10^4^ CFU/g in feces [12], a condition known as super-shedding (SS). A high prevalence of STEC O157 caused by SS events is thought to be a contributing factor to persistence, survival, and infection [13]. Although successful strategies have been implemented to control STEC O157 in food processing [14], on-farm control strategies have been inconsistent [15]. Accordingly, in order to further reduce cross-contamination along the food chain and human disease associated with the consumption of STEC-contaminated food products, effective strategies are required to control STEC O157 shedding by cattle, including SS.

A proper understanding of the mechanisms responsible for SS is necessary, although previous studies of SS have shown variable, and in some cases, conflicting results. The environment, the host, and the pathogen, independently or in combination, are considered to be risk factors responsible for SS [16]. As transmission of STEC O157 is of most concern in slaughter cattle, the role of farm management including stocking density has been evaluated as a risk factor for SS [17]. Others have proposed that the pen within the feedlot is a primary determinant of shedding STEC O157 [18,19], based on the source(s) of cattle within the pen. Alternatively, Robinson et al. [20] proposed that all cattle may become SS at different moments in time, a difficult hypothesis to verify as most studies of STEC O157 in feedlot cattle have cost and logistical restraints precluding frequent or concurrent collection of samples from multiple pens [18]. 

Further defining the factors that are the most important for SS is likely to be complex as these can vary over the year and sampling time [17], and in the same animal over time [20]. Additionally, these factors can be influenced by animal age [21], gut microbiota composition which is diet related, and by other management practices in the feedlot. It has also been suggested that friction in the intestinal walls caused by lower water content of feces can facilitate STEC biofilm detachment and trigger SS [22]. Increased SS has also been associated with higher rainfall and hide contamination [23], and during the first month of lactation, whereby shedding levels tend to decrease with increasing lactating months [17]. Wang et al. [24] suggest that T-cell responses and cholesterol metabolism in the intestinal tract may be associated with SS events, whereas some studies have associated phage type 21/28 with SS strains [25,26].

Genomic polymorphisms are non-synonymous mutations that contribute to the genetic diversity of STEC strains [27]. Genomic mutations range from a single-nucleotide polymorphism (SNP) to large segments of repeat sequences and insertions/deletions. Recently, whole genome sequencing has led to an increase in the number of comparative genomic analyses available to differentiate genetic traits in STEC isolates. For example, Katani et al. [28] revealed differences in SNPs within SS strains compared to other STEC O157 isolates. In the same study, a distinctive pattern of distribution of phage-associated genes was observed amongst the SS compared to other STEC strains. Additionally, Cote, et al. [29] found that an SS isolate differed in the number of SNPs from other STEC strains, but only a single SS isolate was evaluated. In contrast, Munns et al. [30] did not observe genetic differences between SS or low-shedder (LS) isolates and suggested that a cautious approach is required to differentiate SS and LS strains using genomics, as differences in the level of gene expression or genes of unknown function can contribute to STEC O157 shedding. Additionally, STEC strains have been differentiated using the lineage-specific polymorphism assay [31], *stx* prophage integration sites [32], and clade typing [33], which discriminates STEC subpopulations based on their pathogenicity and sources. Characterization of O157 strains by lineage classification and clade profile may also enhance our understanding of risk factors associated with pathogens from a common host.

Hybrid sequence analysis combines short- and long-read sequencing techniques, which produce high-quality assembly, capture most genes, and span repeat and mobile genetic elements in the genome. In this study, we analyzed the genetic profile of STEC O157 strains from cattle from commercial feedlot pens with shedding status determined from a single fecal sample using hybrid sequence analysis to generate an improved assembly, with improved bacteriophage insertion detection [34], and phylogenetics. The intent of this study was to use hybrid sequence analyses to provide a better understanding of SS in order to inform strategies for controlling STEC O157 in the food production system.

## 2. Results

### 2.1. In Silico Detection of Serotype, Multi-Locus Sequence Typing (MLST), Virulence Genes, Insertion Sites, Lineage, and Clade Profile in STEC O157

All isolates were found to carry the H7 antigen determinant (Appendix A). All presumptive SS and LS O157 isolates were ST11 (Table 1). Genes encoding Shiga toxin (*stx*) were found in all STEC O157:H7 genomes. Isolates carried both (*stx1/2a*, n = 58) or either (*stx1a*, n = 4 or *stx2a* n *=* 1). The *stx*-encoding prophage insertion sites *mrlA* and *wrbA* for *stx1a* and *stx2a*, respectively, were occupied, although two isolates had fragments of *stx*-carrying phage sequences in the *mrlA* and *wrbA* loci without *stx1a* and *stx2a*. The plasmid-encoded hemolysin subunit (*ehxA*), the chromosomally encoded intimin (*eae*), the translocated intimin receptor (*tir*), and the anti-terminator *Q933* genes were present in all isolates (Table 1). Additionally, *fimH* and *fumC*, genes involved in cellular adhesion, were also present and similar in all isolates (data not shown).

All the strains screened for lineage-specific polymorphism assay (LSPA-6) were 111111, lineage I (Table 1). Of the 63 isolates, 61 and 2 were identified as clades 1 and 8, respectively. The *stx* subtypes and phage insertion sites of STEC O157 were similar to those of reference strains Sakai and EDL933 (Table 2).

### 2.2. Cluster Analysis between Presumptive Super- and Low-Shedder O157 Strains

Phylogenetic analysis revealed that STEC O157 isolates were distinct from each other and distantly related to most reference strains, although the Sakai reference strain was closely related to strains from a single pen at two feedlots (Figure 1). In addition to the Sakai cluster, other closely related O157 isolates were found across feedlots, including a cluster that contained isolates from each of the three feedlots. Although there were some closely related strains across feedlots, pens within a feedlot were by no means homogenous and there was not a single closely related dominant strain shared by all SS and LS within feedlot pens sharing the same environment, management, and sampling date. 

Additionally, we assembled a phylogenetic tree using SNPs of the isolates in order to see the single-nucleotide differences. The phylogenetic tree (Figure 2) identified 728 SNPs, 44 multiple nucleotide polymorphisms, 47 insertions, and 49 deletions across the evaluated genomes.

Finally, we plotted a graph depicting the pairwise distances between the isolates to facilitate their comparison (Figure 3). Genomic similarity of isolates ranged between 92 and 100%, with all > 96% with the exception of the EDL-933 reference strain.

## 3. Discussion

This study aimed to identify genomic characteristics of STEC O157 isolates from cattle based on shedding status determined at a single sampling to determine the phylogenetic relationships among reference outbreak, SS, and LS strains. STEC O157 isolates were collected from three different commercial feedlots at different times and years and the majority of isolates carried both *stx1a* and *stx2a*. *Stx1a* and *stx2a* are the most dominant *stx* subtypes from cattle and human clinical isolates, respectively, in Alberta [9,10], and *stx2a* is often associated with HUS [39,40]. This emphasizes the importance of these subtypes in bovine sources regarding cross-contamination in the food processing industry. The *stx2a* subtype, which is prophage-encoded, is mostly associated with PT21/28 [41], an important SS phenotype. This subtype was shared in all but four isolated colonies of STEC O157 in the present study. Additionally, all *stx*-encoding prophage insertion sites, *mrlA* and *wrbA* for *stx1a* and *stx2a*, respectively, were occupied, which corroborates other findings [10,42]. However, two isolates had fragments of an *stx*-carrying phage in *mrlA* and *wrbA* loci without *stx1a* and *stx2a*, suggesting that the *stx* gene has been lost, similar to another study [43]. Therefore, prophage regions are potential locations for bacterial genome evolution through loss or acquisition of new genetic material that can alter the virulence of the strain and result in rapid divergence among STEC O157.

Bacterial adherence genes (*eae* and *tir*), plasmid-encoded enterohaemolysin (*ehxA*), and the anti-terminator *Q933* gene, which has been linked to increased toxin production [44], were present in all the isolates. Moreover, the carriage of *eae* indicates that these strains are not only STEC, but they belong to the enterohemorrhagic (EHEC) pathotype. Similarly, *fumC* and *fimH* were present and alleles were identical in all isolates, in contrast to another recent study [45] where these genes were useful for distinguishing clonal sources for on-farm transmission of *E. coli*. These observations, whereby STEC O157 isolates displayed limited variation regarding insertion sites, virulence, and housekeeping gene patterns, support the idea that SS may be an occasional event within cattle that helps to maintain the STEC pool in its bovine reservoir [46], but that it is not strain-dependent. Additionally, the molecular markers evaluated in the present study, some of which were important for virulence of STEC O157, were not sufficient to distinguish SS from LS strains. Similarly, a previous study comparing the genomics of two SS isolates also found them to be highly diverse [28]. Although the virulence repertoire of all SS and LS strains in the present study was similar, additional analyses revealed more genomic diversity.

Different subpopulations of STEC O157 have been grouped into two main lineages, I and II, with lineage I more prevalent among human clinical and bovine sources and lineage II more associated with cattle [31]. All STEC O157 strains in the present study were classified to lineage I, consistent with other studies indicating that lineage I is most prevalent in Alberta and is also associated with bovine sources [9,47], or both clinical and cattle sources [48]. It is thus plausible that persistent isolates from cattle and human origin may be circulating in Alberta, or that cattle are an important source of infection through SS and cross-contamination. Given that environmental conditions such as weather can contribute to SS and that the shedding status of a host can change, sometimes within a single day [16,17,19], associating genomic features with isolates may only assist in identifying closely related strains with similar virulence traits, not necessarily a SS genotype.

STEC O157 strains belonging to clade 8 have been more commonly associated with HUS patients than other clades [33]. Interestingly, 2 of 63 isolates we analyzed were identified as members of clade 8. Although the two isolates, each from a SS or LS source, had similar virulence features to clade 1 isolates, it is apparent that these two isolates are distinct at the genome level based on the clade profile and may offer fitness advantages associated with this clade [49]. The two clade 8 isolates had the *stx2a*-carrying prophage inserted within the *wrbA* insertion site. In contrast, a clade 8 SS strain JEONG-1266 [32] and other clade 8 strains, from a recent study [42], contained the *stx2a*-encoding prophages inserted at *argW* loci. Selection of prophage insertion sites may generate diversity that can be used to further segregate STEC O157 from different geographic regions or environmental sources [50]. However, in the present study, even though isolates were collected from multiple feedlots and over multiple years, all originated within 200 km of each other, demonstrating consistent prophage insertion sites for STEC O157.

Although two of the presumed LS isolates and a single SS isolate subclustered with the Sakai reference strain, phylogenetic analysis for the core genome (Figure 1) revealed that many of the STEC O157 isolates were distinct from each other and only distantly related to other reference strains. Furthermore, this observation is supported by Figure 2 and Figure 3. Despite the SNP phylogenetic tree displaying three major clusters, it did not show a distinct segregation of SS and LS strains. All feedlots sampled were within southern Alberta and the presence of closely related strains across feedlots is evidence of shared sources of cattle and common trucking firms used to transport cattle to and from the feedlots [51]. Nonetheless, Figure 2 illustrates that minor variations among feedlots can be discerned via SNP analysis, as the three clusters tend to group strains from comparable feedlots. This suggests a potential approach for tracing the origins of contaminated food products or pinpointing sources in foodborne outbreaks back to their production locations. Within commercial feedlot pens, which would share the same environment, management, and sampling date, results of this study demonstrated considerable phylogenetic diversity in SS and LS isolates, even though carriage of major virulence factors was consistent. The relative importance of the farm/feedlot and the pen within the feedlot in the transmission of STEC has varied across studies [17,18,19,52]. As the feedlot pens sampled within this study contained up to 350 cattle, genetic relationships among LS and SS isolates of STEC O157 were complex and indicate heterogeneity in the distribution of STEC O157 within the same animal reservoir. Closely related strains were present in pens and within feedlots, but pens with a single dominant strain were limited to those where only one SS was detected. Additionally, the heterogeneity of LS and SS isolated within a pen of cattle demonstrates a lack of influence of SS animals on the carriage of STEC O157 within the pen, in opposition to earlier theories that SS animals were responsible for transmission of 80% of the STEC O157 in the environment [12].

## 4. Conclusions

Conserved genetic trait(s) that can distinguish SS strains were not determined in the present study and may not exist due to the heterogeneity of STEC O157, even in the same feedlot pen. The number of STEC O157 shed may be less important than the ability of these strains to persist in the environment, something that was not possible to determine using the “snapshot in time” approach of the current study necessitated by logistical and budgetary constraints. The STEC O157 isolated within a pen and across feedlots shared the same virulence markers, lineage, and clade profiles as the reference outbreak strains, indicating the threat of these cattle-derived strains to food safety. However, strain differences at the phylogenetic level within the same feedlot pen may increase the difficulty of on-farm STEC control and help to explain the previous variable efficacy of on-farm control measures. Additional studies to evaluate host-related factors influencing SS are warranted.

## 5. Materials and Methods

### 5.1. Strains Selection, DNA Isolation, Genome Sequencing, and Quality Control

Sixty-three STEC O157 strains previously isolated [53,54] in our laboratory from cattle with shedding status based on a single fecal sample collected by digital retrieval (28 LS and 35 SS) were selected for analyses based on detection of SS and LS in the same pen (Appendix A). Isolates were collected from 11 pens in three commercial feedlots in southern Alberta over a seven-year period, from 2007 to 2013. Super-shedders were identified from fecal samples with >10^4^ CFU/g of STEC O157 without enrichment, while LS were detected only after 18 h enrichment using immunomagnetic separation [53]. For resuscitation, strains on glycerol stock (TSB broth and glycerol) at −80 °C were defrosted and reactivated by streaking onto MacConkey agar. A single colony was inoculated into 9 mL of TSB broth over night with shaking. One milliliter of overnight culture was used to extract DNA using the Qiagen Blood and Tissue extraction kit (Qiagen Inc., Toronto, ON, Canada) according to the manufacturer’s instructions. The quality and quantity of the DNA was assessed using Nanodrop and Qubit 2.0 fluorometric systems (Invitrogen Inc., Burlington, ON, Canada), respectively. Samples were sent to Genome Quebec for short-read, 150-base-paired end sequencing (Illumina NovaSeq 6000 platform, Illumina Inc., San Diego, CA, USA; sequencing depth > 300 X.) and the University of Lethbridge for long-read sequencing (PromethION Nanopore system, Nanopore Inc., Montreal, QC, Canada; 8 kpb). The quality of the genomic sequence data was assessed using the FASTQC tool (Version 0.12.1).

### 5.2. Hybrid Genome Assembly

Short read raw data were trimmed using the Trimmomatic tool (version 0.38). Sliding window trimming was conducted with parameters specifying an average across 4 bases, and an average quality threshold set at 20 to remove the Illumina adapters. For generating hybrid assemblies using the Unicycler tool (Version 0.5.0), the pair-end reads from the sequencing of short reads were used along with the Flye-assembled [55] contigs from long-read sequencing. The parameters used were as follows: bridging mode set to normal (moderate contig size and mis-assembly rate), exclusion of contigs from the FASTA file shorter than 100 bp, the lowest k-mer size for SPAdes assembly set at 0.2 times the read length, the highest k-mer size for SPAdes assembly set at 0.95 times the read length, 10 k-mer steps utilized in SPAdes assembly, and filtering out contigs with a depth lower than 0.25 times the chromosomal depth.

The resulting hybrid-assembled contigs of genomes were obtained in FASTA format. The assembled genomes were annotated using Prokka (Version 1.14.6). The parameters employed included a minimum contig size set at 200 bp and a similarity e-value cut-off of 10^6^. The FASTA files were also used for virulence (Abricate Version 1.0.1; *E*. *coli*_VF database), and Multi-locus Sequencing Typing analyses (bio.tools: mlst Version 2.22.0). All assemblies are available in BioProject PRJNA1038721 on the NCBI database.

### 5.3. In Silico Identification of Serogroup, Multilocus Sequence Typing (MLST), Insertion Sites, and Virulence Genes

The EcOH database was used for the determination of the *E. coli* serogroup [56], whereas the seven housekeeping gene loci (*adk*, *fumC*, *gyrB*, *icd*, *mdh*, *purA*, and *recA*) were used for MLST [57]. The O157 *E*. *coli* MLST database provided the sequence types (STs) for STEC isolates. *Stx*-encoding prophage insertion sites were determined as previously described [10]. The 1311 bp gene sequence of *stx2*-carrying prophage integrase inserted at the NADH quinone oxidoreductase (*wrbA*) loci (579 bp) in the Sakai strain (NC002695) was extracted and searched for against O157 strains using Geneious prime 2023 (https://www/geneious.com (accessed on 5 July 2023)). The flanking loci were identified as *wrbA* in O157 isolates. Similarly, a 1287 bp *stx1*-carrying prophage integrase site flanking the transcriptional activator (*mlrA*) (648 bp) insertion site in Sakai was obtained and verified against the O157 isolates. The presence of an integrase–insertion site combination was defined as an occupied site. The *stx* gene in the occupied prophage sequence was further verified by a subsequent sequence search using Geneious. The absence of a *stx* gene in the prophage sequence or in the entire genome was classified as loss of the *stx* gene. As *mlrA* and *wrbA* insertion sites were all occupied, other insertion sites such as *argW*, *yecE*, and *sbcB* were not further characterized. For virulence analysis, we used the Virulence Finder database with default settings: coverage ≥ 60% and identity ≥ 90% (https://cge.cbs.dtu.dk/services/VirulenceFinder, accessed on 28 August 2023) [58].

### 5.4. In Silico Detection of Anti-Terminator Q933 Gene, Lineage-Specific Polymorphism Assay (LSPA-6), and Clade Genotyping using Geneious Prime

The genome sequence of O157 isolates was screened for the anti-terminator Q933 gene using the primer sequence [44]. The anti-terminator *Q* genes located downstream and upstream of the *stx1* and *stx2* genes, respectively, were considered positive. LSPA-6 analyses were performed using six pairs of primers that target the genomic loci *fold-sfmA*, *Z5935*, *yhcG*, *rtcB*, *rbsB*, and *arp-iclR* [31]. A six-digit binary code was used to determine strains with LSPA-6 profile 111111 as lineage I. Clades 1 and 8 were identified using two primer combinations *rhsA-C* plus *rhsA*-Sakai, and *rhsA*-C plus *rhsA*-8, respectively [59,60]. A 578 bp gene sequence was considered positive for both clades. Reference strains Sakai (NC002695) and JEONG-1266 (CP014314; a super-shedder bovine strain) [37] were used as representative strains for clades 1 and 8, respectively.

### 5.5. Cluster Analysis between Presumptive Super- and Low-Shedder O157 Strains

After obtaining the annotated data through Prokka, we performed a core-genome alignment using gff3 format files. Thus, we selected all annotated hybrids and used the Roary tool (Version 3.13.0) to create the core genome. For this, we used a minimum percentage identity of 95% and a percentage of core-gene isolates of 10%. The parameters utilized were a maximum number of clusters set at 50,000. The core-genome alignment was obtained in FASTA format, and the file was used to build a phylogenetic tree to understand the genetic relationship between the isolates. Therefore, we used the IQ-Tree to build a genetic tree of the relationship between SS and LS with an ultrafast bootstrap value (n = 1000). The .nhx file was used for better visualization of the genetic tree using ITOL version 6.8.1. TW14359; NC013008.1 [36], EC4115.1; NC011353.1 [38], Sakai; NC002695.2 [6], EDL933; CP008957.1 [35] and SS17; CP008805.1 [29], SS52; CP010304.1; [28], JEONG-1266; CP014314 [37], which are outbreak and SS strains, were used as reference strains.

### 5.6. SNP Tree and Pairwise Comparison

We conducted a phylogenetic analysis using the SNIPPY software (Version 4.6.0) to pinpoint genomic variations. Our reference for mapping the isolates was *E. coli* O157:H7 strain EDL-933 (GCA_000732965.1). Parameters used were: Minimum mapping quality = 60; Minimum coverage = 10; Minimum proportion for variant evidence = 0.9. The SNIPPY direction file’s output was employed to create the SNIPPY-CORE, an alignment of the SNPs obtained across multiple sequences. This data was then utilized to construct the phylogenetic tree using the IQ-Tree software (Version 2.1.2), applying a bootstrap value of 1000 for robustness. The resulting tree visualization was refined using the ITOL version 6.8.1 software.

For the pairwise genomic comparison, we employed the SNP distance matrix pipeline (snp dists, Version 0.8.2), which enabled us to calculate distances between each SNP in the isolates and to generate a corresponding data matrix. We then analyzed this matrix using the ggplot2 tool.

## Figures and Tables

**Figure 1 toxins-16-00086-f001:**
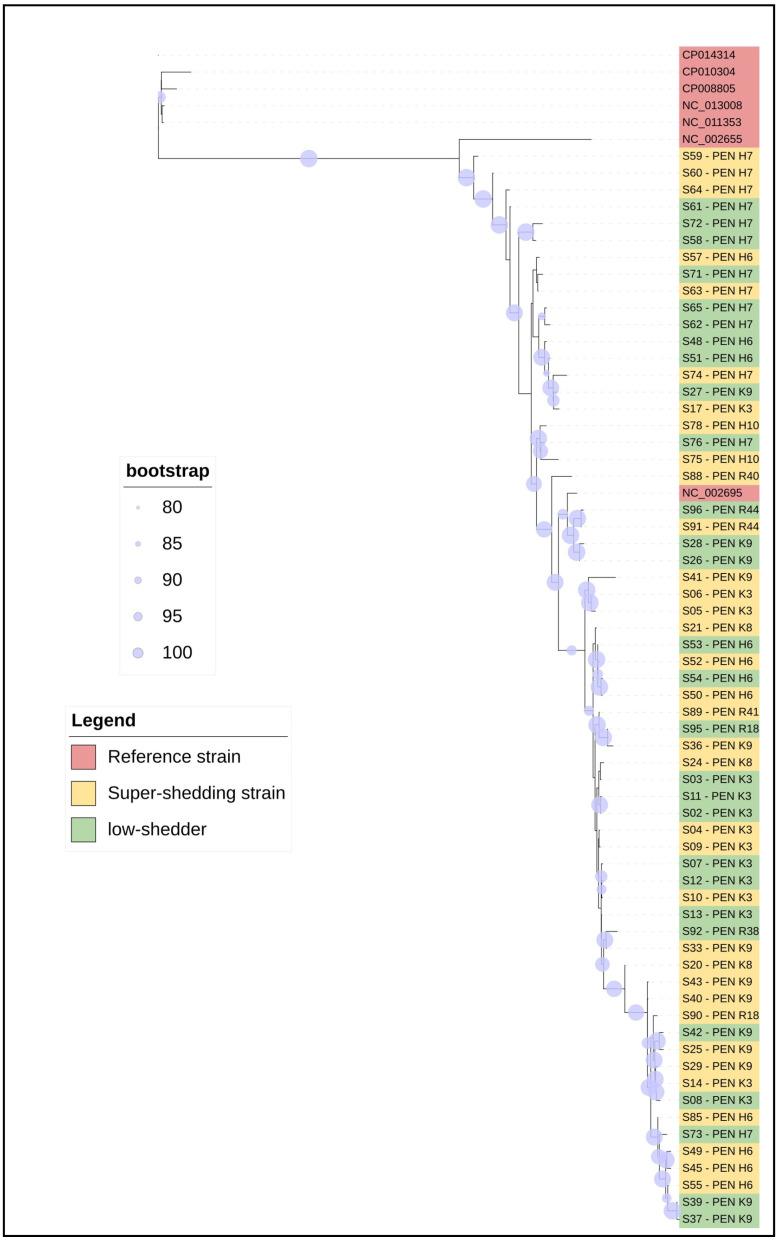
The core genome tree is shown. STEC O157 isolates belonging to super shedder, low shedder, and reference strains are listed in green, yellow, and red, respectively.

**Figure 2 toxins-16-00086-f002:**
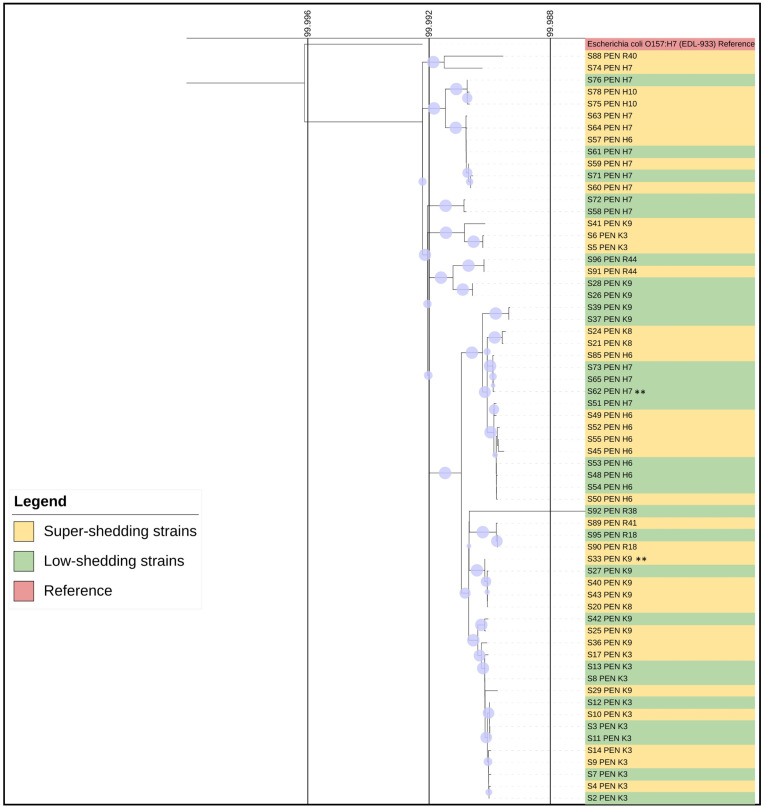
SNP phylogenetic tree. Isolates of STEC O157 associated with super shedders, low shedders, and reference strains are color-coded in green, yellow, and red, respectively, for easy identification. Strains belonging to clade 8 are distinctly marked with a double asterisk. The scale provided illustrates the level of similarity among the isolates, quantifying it as a percentage to denote the extent of genetic resemblance. Additionally, the bootstrap values, which are above 80%, are indicated by circles at each leaf node of the phylogenetic tree, emphasizing the statistical confidence in the tree’s branching structure.

**Figure 3 toxins-16-00086-f003:**
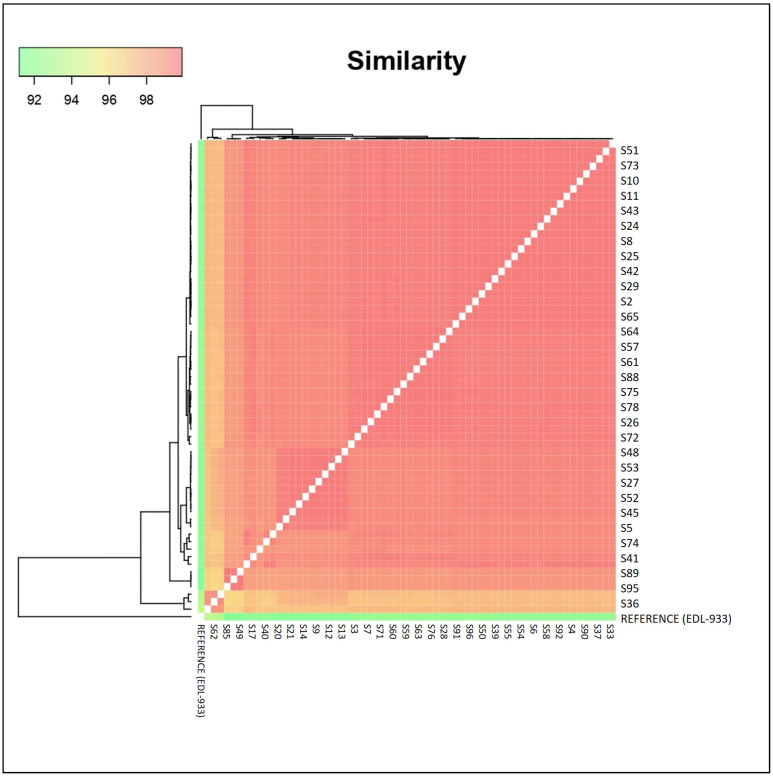
A pairwise comparison of genomic similarity among the isolates. The intensity of the red color indicates a greater genomic closeness between the isolates.

**Table 1 toxins-16-00086-t001:** Distribution of virulence, lineage, clade profile, insertion sites, and other genes among STEC O157:H7 isolates.

Number of Isolates	Source	MLST	Lineage	Clade	Insertion Site		*stx* Subtype	*eae*	*tir*	*ehxA*	*Q933*
33	SS	11	I	1	*mlrA*	*wrbA*	*Stx1a/2a*	+	+	+	+
1	SS			8	*mlrA*	*wrbA*	*Stx 1a/2a*	+	+	+	+
1	SS			1	*mlrA*	-	*Stx 1a*	+	+	+	+
24	LS			1	*mlrA*	*wrbA*	*Stx 1a/2a*	+	+	+	+
1	LS			8	*mlrA*	*wrbA*	*Stx 1a/2a*	+	+	+	+
1	LS			1	*mlrA*	-	*Stx 1a*	+	+	+	+
1	LS			1	*mlrA* *	*wrbA*	*Stx 2a*	+	+	+	+
1	LS			1	*mlrA*	*wrbA* *	*Stx 1a*	+	+	+	+

SS; super shedding, LS; low shedding, *; locus present but stx gene absent, +; gene present, -; gene absent.

**Table 2 toxins-16-00086-t002:** Distribution of lineage, clade profile, *stx* subtype, and insertion sites of reference O157 strains.

Strain ID	Source	Lineage	Clade	stx Subtype	Insertion Site		Accession Number	Year Isolated	References
Sakai	Clinical	I	1	*Stx1a/2a*	*mlrA*	*wrbA*	NC002695.2	1996	[6]
EDL933	Ground beef	I	3	*Stx1a/2a*	*mlrA*	*wrbA*	CP008957.1	1982	[35]
TW14359	Clinical	I/II	8	*Stx2a/2c*	-	*argW*	NC013008.1	2006	[36]
JEONG-1266	Cattle	I/II	8	*Stx2a/2c*	-	*argW*	CP014314	2011	[37]
SS17	Cattle	I/II	8	*Stx2a/2c*	-	*argW*	CP008805.1	2009/10	[29]
SS52	Cattle	I/II	-	-	-	*-*	CP010304.1	2009/10	[28]
EC4115.1	Spinach	I/II	8	*Stx2a/2c*	-	*argW*	NC011353.1	2006	[38]

## Data Availability

Data is available in BioProject PRJNA1038721 on the NCBI database.

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
