# Peer review of "In Silico Analysis of Shiga Toxin-Producing *Escherichia coli* O157:H7 Strains from Presumptive Super- and Low-Shedder Cattle"

_toxins, 2024, doi:10.3390/toxins16020086_

Round 1

Reviewer 1 Report

Comments and Suggestions for Authors

The manuscript is well written and informative.

Author Response

Thank-you Reviewer #1.  Glad you liked it.

Reviewer 2 Report

Comments and Suggestions for Authors

The manuscript contains an excellently written description of an important bacterial genomic project aiming to explore the genomic background of the supershedding phenomenon of E. coli O157 strains, and seems to be a continuation of previous efforts which also yielded valuable results.

The limitation of the study was the 'snapshot' nature of the sampling, but this was adequately addressed by the authors. Nevertheless, a continuous monitoring of even a single cattle farm could yield valuable information on the time-related features of supershedding.

The Introduction meticulously covers every aspect of the problem so far, the design of the study is adequate, the sequencing and bioinformatics were all done using cutting edge tools. The results are presented clearly, and the Discussion is prudent. I wonder if some of the non-core genome elements, which are not directly related to virulence, could be important factors in the SS phenomenon? The authors rightly point out the importance of prophages in genome evolution, but it just occurred to me if there is some hitherto unidentified prophage-based factor behind SS?

There are two small remarks regarding notions in the Discussion: 

L145-146: 'However, two isolates had occupied mrlA and wrbA loci without stx1a and 145

stx2a, indicating that the stx gene has been lost [43].'

I have two questions regarding this statement:

1. it is not clear, what the reference stands for. If it indicates a similar case (where the typical stx prophage integration sites were occupied by an other insert in a stx-negative isolate), the sentence should indicate this more explicitly (e. g. 'similarly to another study').

2. I am not really sure whether the alternative insert accounts for a 'loss' of the stx prophage. It maybe possible that these strains never carried stx prophages to begin with?...

L152: I would add that the carriage of eae indicates that these strains are not only STEC, but they belong to the enterohemorrhagic (EHEC) pathotype

Author Response

The manuscript contains an excellently written description of an important bacterial genomic project aiming to explore the genomic background of the supershedding phenomenon of E. coli O157 strains, and seems to be a continuation of previous efforts which also yielded valuable results.

The limitation of the study was the 'snapshot' nature of the sampling, but this was adequately addressed by the authors. Nevertheless, a continuous monitoring of even a single cattle farm could yield valuable information on the time-related features of supershedding.

The Introduction meticulously covers every aspect of the problem so far, the design of the study is adequate, the sequencing and bioinformatics were all done using cutting edge tools. The results are presented clearly, and the Discussion is prudent. I wonder if some of the non-core genome elements, which are not directly related to virulence, could be important factors in the SS phenomenon? The authors rightly point out the importance of prophages in genome evolution, but it just occurred to me if there is some hitherto unidentified prophage-based factor behind SS?

There are two small remarks regarding notions in the Discussion:

L145-146: 'However, two isolates had occupied mrlA and wrbA loci without stx1a and 145

stx2a, indicating that the stx gene has been lost [43].

I have two questions regarding this statement:

1: it is not clear, what the reference stands for. If it indicates a similar case (where the typical stx prophage integration sites were occupied by an other insert in a stx-negative isolate), the sentence should indicate this more explicitly (e. g. 'similarly to another study').

Response: The sentence has been rephrased as requested line 154-147.

2: I am not really sure whether the alternative insert accounts for a 'loss' of the stx prophage. It maybe possible that these strains never carried stx prophages to begin with?

Response: The strains had fragments (remnants) of stx-carrying prophage sequences in their genome. Sentence has been rephrased in line 145.

L152: I would add that the carriage of eae indicates that these strains are not only STEC, but they belong to the enterohemorrhagic (EHEC) pathotype

Response: The sentence has been modified as suggested in line 152.

Reviewer 3 Report

Comments and Suggestions for Authors

This manuscript describes a genomic comparison of 63 archived E. coli 0157 isolates previously determined to be from super shedder (SS) or low shedder (LS) cattle.  Isolates were from different commercial feed lots, sampled at varied times over 7 years, and were sampled to provide snapshots of the array of detectable SS and LS genotypes, to better understand O157 super shedding and hopefully inform control strategies. 

The manuscript has a well written and referenced introduction, providing important background on O157 super shedding.  Hybrid long read/short sequencing reads were used to enhance the comparison.

All isolates had the same MLST and LSPA-6 types, and all were found harbor known virulence genes: eae, tir, ehxA and Q933, but there was some variation in their shiga toxin (stx) gene content. The majority (59/63) harbored both stx1a and stx2a, and of these 34 (58%) were SS and 25 (42%) were LS, suggesting that other factors were important in dictating shedding status. 

The phylogenomic analysis revealed that although there was apparent clustering of isolates within feedlots, feedlots and feed lot pens were diverse and there was no single closely related strain shared by all animals within a feedlot pen.  The observed diversity will make control based on isolate genotype potentially challenging. 

There were several instances where members of the same cluster groups were isolated from different feedlots, presumably from shared sources of cattle and or cattle transport.  

There were numerous instances where close relatives had different shedding status. 

The authors conclude that based on their analysis there was no obvious genetic traits associated with SS.  These traits may be difficult to decipher because of the other factors (host, environment) that influence supper shedding. 

Please provide clarification:

·         The text describes samples coming from 3 feed lots (line 11, line 136) although only two are listed as sources in supplementary Table 1. 

·         There are additional samples shown in the phylogenetic tree- (they have pen prefix R).  Please indicate why they are not listed in supplementary table 1.

·         Line 100-101- would be clearer if “subtypes 1a and 2a” was left out.

·         There is no indication of the quality of the genomic data.  Average coverage for short read and assembly statistics for long read and hybrid assemblies. Were these all completely closed genomes? if not how many contigs?

·         Does the phylogeny compare SNPs as well as INDELs? 

·         Please provide a diversity scale for the tree.

Please indicate the year of isolation for the examined samples and the standards on the phylogenetic tree, as this will greatly aid in interpreting the genomic relationships.

Please indicate in the phylogeny which of the isolates were “clade 8”.  Do they cluster together on the tree?

A pairwise distance plot would greatly improve the ability to compare the genomic similarity of these isolates. 

This manuscript would be improved by providing a list of non-synonymous mutations to see if there were correlations between SS and changes in specific genes, operons or ontological classes.

Author Response

This manuscript describes a genomic comparison of 63 archived E. coli 0157 isolates previously determined to be from super shedder (SS) or low shedder (LS) cattle.  Isolates were from different commercial feed lots, sampled at varied times over 7 years, and were sampled to provide snapshots of the array of detectable SS and LS genotypes, to better understand O157 super shedding and hopefully inform control strategies. 

The manuscript has a well written and referenced introduction, providing important background on O157 super shedding.  Hybrid long read/short sequencing reads were used to enhance the comparison.

All isolates had the same MLST and LSPA-6 types, and all were found to harbor known virulence genes: eae, tir, ehxA and Q933, but there was some variation in their Shiga toxin (stx) gene content. The majority (59/63) harbored both stx1a and stx2a, and of these 34 (58%) were SS and 25 (42%) were LS, suggesting that other factors were important in dictating shedding status. 

The phylogenomic analysis revealed that although there was apparent clustering of isolates within feedlots, feedlots and feed lot pens were diverse and there was no single closely related strain shared by all animals within a feedlot pen.  The observed diversity will make control based on isolate genotype potentially challenging. 

There were several instances where members of the same cluster groups were isolated from different feedlots, presumably from shared sources of cattle and or cattle transport. 

There were numerous instances where close relatives had different shedding status. 

The authors conclude that based on their analysis there was no obvious genetic traits associated with SS.  These traits may be difficult to decipher because of the other factors (host, environment) that influence supper shedding. 

Please provide clarification:

1: The text describes samples coming from 3 feed lots (line 11, line 136) although only two are listed as sources in supplementary Table 1.

Response: Three feedlots, ABC are indicated in the supplementary Table 1.

2: There are additional samples shown in the phylogenetic tree- (they have pen prefix R).  Please indicate why they are not listed in supplementary table 1.

Response: These are reference strains from the NCBI database, and their accession numbers are included in text for perusal.

3: Line 100-101- would be clearer if “subtypes 1a and 2a” was left out.

Response: The wording has been altered as suggested.

  1. There is no indication of the quality of the genomic data. Average coverage for short read and assembly statistics for long read and hybrid assemblies. Were these all completely closed genomes? if not how many contigs?

Good point. We have included a new table as a supplementary file with the quality data for the sequences. Thank you for the suggestion.

  1. Does the phylogeny compare SNPs as well as INDELs?

R: We have incorporated a new tree depicting the SNPs and have detailed the obtained SNP and INDEL data in the legend for clarity.  INDELs are shown in the new Supplementary Table 3.

  1. Please provide a diversity scale for the tree.

R: A diversity scale was provided in the new Figure 2. Thank you for the suggestion.

  1. Please indicate the year of isolation for the examined samples and the standards on the phylogenetic tree, as this will greatly aid in interpreting the genomic relationships.

R:  We have modified Supplementary Table 1 to include the year of isolation and also included year of isolation of standards on Table 2.  Was a bit messy when we tried to add year of isolation also on Figure 1.  

  1. Please indicate in the phylogeny which of the isolates were “clade 8”. Do they cluster together on the tree?

R: In Figure 2, we marked the two samples from Clade 8, and they were grouped into two distinct clusters.

  1. A pairwise distance plot would greatly improve the ability to compare the genomic similarity of these isolates.

R: We have included a pairwise distance plot between the isolates to facilitate the analysis.  This is now Figure 3. Thank you for the suggestion.